# A Geometric Observer-Assisted Approach to Tailor State Estimation in a Bioreactor for Ethanol Production

Silvia Lisci, Massimiliano Grosso and Stefania Tronci *

Dipartimento di Ingegneria Meccanica Chimica e dei Materiali, Università degli Studi di Cagliari, Via Marengo 2, 09123 Cagliari, Italy; s.lisci@dimcm.unica.it (S.L.); massimiliano.grosso@dimcm.unica.it (M.G.)
* Correspondence: stefania.tronci@dimcm.unica.it; Tel.: +39-070-675-5050

**Abstract:** In this work, a systematic approach based on the geometric observer is proposed to design a model-based soft sensor, which allows the estimation of quality indexes in a bioreactor. The study is focused on the structure design problem where the set of innovated states has to be chosen. On the basis of robust exponential estimability arguments, it is found that it is possible to distinguish all the unmeasured states if temperature and dissolved oxygen concentration measurements are combined with substrate concentrations. The proposed estimator structure is then validated through numerical simulation considering two different measurement processor algorithms: the geometric observer and the extended Kalman filter.

**Keywords:** nonlinear state estimation; geometric observer; bioreactor; continuous system; extended Kalman filter; model-based sensor

---

## 1. Introduction

Bioreactors are units where a wide variety of products are made in industrial plants and where a diversity of important processes, such as fermentation, occur. Usually, the control of bioreactors is accomplished through the regulation of variables, such as pH and temperature, for optimizing the microbial growth [1,2]. Product quality indexes such as biomass, substrate, product or by-product, and dissolved oxygen concentrations are not usually controlled, because they are difficult to measure in real-time [3]. Even though many works report the availability and advantages of monitoring techniques, industrial biotechnology processes have a scarce capacity for real-time monitoring, which implies a limited implementation of efficient control of the process [4,5]. Because an unpredicted perturbation may lead to significant changes in the qualitative behavior of the system [6–8], it is crucial to accurately monitor the process.

Model-driven soft sensors can be a possible approach to estimate variables from secondary measurements. They rely on first principles process models and on algorithms that reconcile the available measurements with predictions carried out by the model. Several estimation techniques have been proposed in the literature for chemical and biochemical processes. Among them, the following have been recognized to have strong potential in the online estimation of nonlinear systems: (i) extended Kalman filter [9], (ii) high gain observer [10], (iii) sliding mode observer [11], (iv) geometric observer [12,13]. Many of the strategies to estimate unmeasurable states and disturbances for partially known systems are based on the extended Kalman filter (EKF) because its design is quite simple and it is widely accepted by relevant industries [14,15].

In this paper, the problem of estimating unmeasured states in a bioreactor is addressed. The study is based on the detailed model proposed by [16], which is considered as the virtual plant. The main

objective is to compare different estimation solutions depending on the available measurements and the characteristics of the sensors. An adjustable-structure geometric estimation approach is implemented, considering the estimator structure as a degree of freedom in the design with the aim of improving performance versus robustness estimation behavior [13,17,18]. The used estimation algorithm is the geometric observer (GO) with proportional innovation [19], which offers the simplicity of tuning and implementation. In order to show that the proposed procedure for choosing the estimation structure can be applied to other estimation techniques, the extended Kalman filter (EKF) is also used as the measurement processor algorithm.

## 2. Process Model

The biochemical process considered in the present paper is a fermentation reactor for the production of ethanol. The model was developed by [16] and, for the sake of clarity, it is hereafter reported (Equations (1)–(6)). It is assumed there is a perfect mixing in the reactor (constant pH and constant volume). The dynamics of biomass ($C_X$), substrate ($C_S$), ethanol ($C_P$), along with the oxygen concentration ($C_{O_2}$) are considered. Energy balances are also taken into account describing the reactor temperature ($T_r$) and jacket temperature ($T_{ag}$) dynamics. A low dilution rate has been considered allowing a balance between the biomass exiting from the system and the biomass produced in the reactor.

$$\frac{dC_X}{dt} = \mu_X \, C_X \, \frac{C_S}{K_S + C_S} e^{-K_P C_P} - \frac{F_e}{V} C_X \tag{1}$$

$$\frac{dC_P}{dt} = \mu_P C_X \frac{C_S}{K_{S1} + C_S} e^{-K_{P1} \, c_p} - \frac{F_e}{V} \, C_P \tag{2}$$

$$\frac{dC_S}{dt} = -\frac{1}{R_{SX}} \mu_X C_X \frac{C_S}{K_S + C_S} e^{-K_P C_P} - \frac{1}{R_{SP}} \mu_P C_X \frac{C_S}{K_{S1} + C_S} \, e^{-K_{P1} C_P} + \frac{F_i}{V} C_{S,in} - \frac{F_e}{V} C_S \tag{3}$$

$$\frac{dC_{O_2}}{dt} = k_l a \left( C_{O_2}^* - C_{O_2} \right) - \mu_{O_2} \frac{1}{Y_{O_2}} C_X \frac{C_{O_2}}{K_{O_2} + C_{O_2}} \tag{4}$$

$$\frac{dT_r}{dt} = \left( \frac{F_i}{V} \right)(T_{in} + 273) - \left( \frac{F_e}{V} \right)(T_r + 273) - \mu_{O_2} \frac{1}{Y_{O_2}} C_X \frac{C_{O_2}}{K_{O_2} + C_{O_2}} \frac{\Delta H_r}{32 \, \rho_r \, C_{heat,r}} - \frac{K_T A_T \left( T_r - T_{ag} \right)}{V \, \rho_r \, C_{heat,r}} \tag{5}$$

$$\frac{dT_{ag}}{dt} = \left( \frac{F_{ag}}{V_j} \right)\left( T_{in,ag} - T_{ag} \right) + \frac{K_T A_T \left( T_r - T_{ag} \right)}{V_j \rho_{ag} C_{heat,ag}} \tag{6}$$

The oxygen equilibrium concentration is affected by the inorganic salts, which are added to the solution as source of inorganic nitrogen. The dependence is reported in Equation (7)

$$C_{O_2}^* = C_{O_2, \, 0}^* \cdot 10^{-\sum H_i I_i} \tag{7}$$

where the equilibrium concentration as a function of temperature has been calculated using the equation proposed by [20] for distilled water

$$C_{O_2, \, 0}^* = 14.6 - 0.3943 T_r + 0.007714 \, T_r^2 - 0.0000646 T_r^3 \tag{8}$$

In the present work, the distillation strength $\sum H_i I_i$ is kept constant and it has been calculated using the equations reported in [16].

The model is here used to simulate a real process and to develop the model-based soft sensor (estimator). Because the aim of the work is to mimic a real situation, the simulation using the model parameters reported in [16] is considered as the real plant (hereafter referred to as the virtual plant). On the other hand, some of the parameters used for the model in the estimator algorithm have been modified. The aim was to insert modeling errors to simulate what usually happens in a real situation where parameter uncertainty is present. This is often the case when dealing with complex systems such

as biological reactors [21]. A Monte Carlo method has been used to produce empirical error estimates on the model parameters, using a uniform noise distribution with a maximum deviation equal to ±8%. Table 1 summarizes the parameters of the model used in the estimator algorithm obtained by performing 100 simulations and leading to the maximum error calculated on the trajectories of the six states. The other parameters of the model are the same as reported in [16]. The nominal conditions of the virtual plant are reported in Table 2.

**Table 1.** Parameters for the virtual plant and for the estimator.

|  | Virtual Plant | Estimator |
|---|---|---|
| $\mu_p$ | 1.790 | 1.7465 |
| $K_s$ | 1.030 | 1.0248 |
| $K_p$ | 0.139 | 0.1281 |
| $K_{s1}$ | 1.680 | 1.8090 |
| $K_{p1}$ | 0.070 | 0.0692 |
| $R_{sx}$ | 0.607 | 0.6274 |
| $R_{sp}$ | 0.435 | 0.4549 |

**Table 2.** Nominal operating conditions of the process.

| $C_{O2}$ = 2.5 mg/L | pH = 6 |
|---|---|
| $C_P$ = 13 g/L | $T_{ag}$ = 29 °C |
| $C_S$ = 27 g/L | $T_{in}$ = 25 °C |
| $C_{S,in}$ = 60 g/L | $T_{in,ag}$ = 15 °C |
| $C_X$ = 1 g/L | $T_r$ = 26 °C |
| $F_{ag}$ = 18 L/h | $V$ = 1000 L |
| $F_i = F_e$ = 51 L/h | - |

The model simulations have also been made more realistic by adding noise to the available measurements, and the precision of the sensors is reported in Table 3.

**Table 3.** Noise for the different measuring sensors.

| $C_X$ | $C_S$ | $C_{O2}$ | $T_r$ | $T_{ag}$ |
|---|---|---|---|---|
| ±2.5% | ±2.5% | ±2.5% | ±0.1 °C | ±0.1 °C |

It is important to specify that all tests performed in the following have been carried out by imposing step changes to three inputs of the system ($C_{S,in}$, $T_{in,ag}$, $T_{in}$). The description of input variations is reported in Table 4.

**Table 4.** Step changes of input variables.

|  | Input | $t$ = 0 h | $t$ = 100 h | $t$ = 200 h |
|---|---|---|---|---|
| T1 | $C_{S,in}$ (g/L) | 60 | 45 | 75 |
| T2 | $T_{in,ag}$ (°C) | 15 | 10 | 20 |
| T3 | $T_{in}$ (°C) | 25 | 20 | 30 |

## 3. Estimation Problem

The current real-time monitoring methods used in ethanol production consist of secondary measurements such as pH, turbidity, gas composition and temperature [4]. Even if such variables provide important information about the process, they do not directly relate to the state of the system, making it difficult to apply advanced control strategies. Furthermore, even the best process measurements are corrupted by some amount of signal noise and their true values are somewhat uncertain. State estimation techniques can be used to improve the output signal of measured process states in the presence of uncertainty and when it is not possible to directly measure all the variables of interest.

The estimation problem consists of jointly designing the estimation structure (i.e., estimator model, sensors, innovated states and data assimilation mechanisms), and the estimation algorithm (i.e., the dynamic data processor), to infer some or all the states of the bioreactor on the basis of the available model in conjunction with available measurements, according to a specific estimation objective. In the present fermentation reactor estimation study, the emphasis has been placed on: (i) the detection of the more adequate measured outputs leading to the best performance, (ii) the selection of the innovated states, meaning the states which are updated by using the available measurement.

For simplifying the formulation of the problem, the model in Equations (1)–(6) is written in compact form as reported in Equation (9)

$$\dot{x} = f(x, u), \quad x(t_0) = x_0 \tag{9a}$$

$$y = h(x) \tag{9b}$$

where $x$ is the $n$-dimensional state vector, equal to $x_0$ at the initial time $t_0$, $u$ is the $p$-dimensional input vector, $f$ is the n-dimensional vector fields, $y$ is the $m$–dimensional vector of the measured outputs and $h$ is the map relating states and measurements. The dimension of the measured outputs is less than the number of states, that is $m < n$. Using the geometric approach [19,22], it is possible to define the nonlinear estimation map $\phi$ as Equation (10)

$$\phi(x, u) = [\phi_1, \ldots, \phi_i, \ldots, \phi_m]^T \tag{10a}$$

$$\phi_i = \left( h_i, \ L_f h_i, \ \ldots, L_f^{\kappa_i - 1} h_i \right) \tag{10b}$$

where the $L_f^j h_i$ is the $j$th Lie derivative of the time-varying scalar field $h_i$ along the vector $f$, $\kappa_i$ is the observability index of the $i$th output and $\kappa$ is the estimator order defined in Equation (11)

$$\kappa_1 + \kappa_2 + \ldots + \kappa_m = \kappa = n \tag{11}$$

If the map $\phi(x, u)$ is invertible with respect to $x$ (Equation (12)), the system is observable, and the states can be reconstructed using the available model (Equations (1)–(6)) and a proper measurement processor algorithm [14].

$$rank(\partial_x \phi(x, u)) = n \tag{12}$$

If the Jacobian matrix $\partial_x \phi(x, u)$ is rank deficient, there are unobservable states. In this case, the system is detectable only if all the unobservable modes have negative real parts [19].

### 3.1. Robust Estimability and Robust Detectability

If all states can be fully observable, the observability matrix should be full-rank, but practical observability can be assessed if the condition number of the observability matrix ($\Sigma$) is small [23].

Furthermore, a small singular value of the observability matrix implies the worst estimate of the states [24]

$$rank(\mathbf{\Upsilon}) = n, \mathbf{\Upsilon} = \partial_x \mathbf{\Phi}(x, u) \tag{13a}$$

$$\frac{\overline{\sigma}(\mathbf{\Upsilon})}{\underline{\sigma}(\mathbf{\Upsilon})} = \Sigma < \Xi, \tag{13b}$$

$$avg\underline{\sigma}_t(\mathbf{\Upsilon}) \geq \varepsilon_0 \tag{13c}$$

where $\Xi$ and $\varepsilon_0$ are, respectively, the selected thresholds.

On the other hand, if matrix $\mathbf{\Upsilon}$ is rank deficient and the unobservable states are stable, it is necessary to distinguish between states that can be innovated (distinguishable states) and states that cannot (undistinguishable states). In this case, the dimension of the map in Equation (10) is equal to the dimension of the distinguishable states, and robust detectability can be assessed if the following conditions are satisfied (Equation (14))

$$\frac{\overline{\sigma}(\mathbf{\Upsilon}_p)}{\underline{\sigma}(\mathbf{\Upsilon}_p)} = \Sigma_i < \Xi_p \tag{14a}$$

$$avg\underline{\sigma}_t(\mathbf{\Upsilon}_p) \geq \varepsilon_{p0} \tag{14b}$$

$$\mathbf{\Upsilon}_p = \partial_x \mathbf{\Phi}_p(x, u) \tag{14c}$$

$$\mathbf{\Phi}_p = \left( h_1, \ldots, L_f^{\kappa_1 - 1} h_1, \ldots, h_m, \ldots, L_f^{\kappa_m - 1} h_m \right) \tag{14d}$$

$$\kappa_1 + \kappa_2 + \ldots + \kappa_m = \kappa = p \tag{14e}$$

The constants $\Xi_p$ and $\varepsilon_{p0}$ are, again, the selected thresholds.

### 3.2. Selection of the Estimator Structure

The performance of an estimator is obviously strongly affected by the model of the process and the quality of the available measurements. Biological processes are complex systems, therefore the presence of model uncertainty in terms of parameters and neglected dynamics is, in general, to be expected [21]. This means that the complete reconstruction of the states requires, in general, a combination of different measurements [4]. Within this framework, it is important to underline that there is still a gap between the sensors for laboratory use and large scale monitoring in real-time [4]. The selection of the estimator structure is therefore focused on the choice of the best monitoring strategies, by considering which are the most representative measured outputs and the presence of parameter errors in the model used in the estimator. It is considered that system monitoring can be expensive, in terms of both fixed and operation costs, therefore it could be useful to optimize performance with the least number of sensors. This analysis has been carried out comparing condition number and minimum singular values of the matrix $\mathbf{\Upsilon}$ or $\mathbf{\Upsilon}_p$ (Equations (13) and (14)). The performances have also been evaluated by simulating different trajectories, from which the convergence rate, presence of off-set and signal noise have been evaluated.

### 3.3. Algorithms for the Estimation Problem

In this study, two different algorithms have been selected and compared. The first one is the geometric observer [19], which is formally connected with the observability properties reported in the previous section. The geometric observer (GO) can be applied also to detectable systems [13], and it demonstrated to be simple to implement and tune [18,22]. The geometric approach is also used to select the estimator structure, in terms of the selection of measurements and states to be innovated.

The geometric observer algorithm is reported in Equation (15), where it is assumed that some states are not innovated. This choice may depend on the rank deficiency of the observability matrix or a design choice intended to improve the robustness and efficiency of the estimator.

$$\dot{\hat{x}}_i = \hat{f}_i(\hat{x}, u) + (\partial_{xi}\phi(\hat{x}, u))^{-1} K(y - h(\hat{x})), x_{i0} = x_i(t_0) \tag{15a}$$

$$\dot{\hat{x}}_u = \hat{f}_u(\hat{x}, u), x_{u0} = x_u(t_0) \tag{15b}$$

The inverse of the observability matrix $\partial_{x_i}\phi(x, u)$ in Equation (15a) is calculated at each time step and $K$ is a block diagonal matrix (Equation (16)), whose coefficients are constant tuning parameters. The estimated states $\hat{x}$ in Equation (15) are the innovated states ($\hat{x}_i$), where the dynamics predicted by the model are adjusted by means of the available measurements $y$, and the not innovated states ($\hat{x}_u$), which are only predicted by the model (referred to in the following as open-loop states).

$$K = \begin{pmatrix} B_1 & 0 & \dots & 0 \\ 0 & B_2 & \dots & 0 \\ \vdots & \vdots & \dots & \vdots \\ 0 & 0 & \dots & B_m \end{pmatrix}, \quad B_1 = \begin{bmatrix} k_{11} \\ \vdots \\ k_{1v_1} \end{bmatrix}, \quad B_2 = \begin{bmatrix} k_{21} \\ \vdots \\ k_{2v_2} \end{bmatrix}, \quad B_m = \begin{bmatrix} k_{m1} \\ \vdots \\ k_{mv_m} \end{bmatrix} \tag{16}$$

$$v_i = \kappa_{i-1}$$

Tuning guidelines are provided by [17], proving that a set of tuning parameters $k_{ij}$ is required for every measurement. For observability indexes equal to 1 or 2 ($\kappa_i = 1, 2$), the proportional gains can be obtained by considering Equation (17).

$$k_{i1} = 2\zeta\omega_0, \quad k_{i2} = \omega_0{}^2 \tag{17a}$$

$$\omega_0 \in [10\omega_c, 30\omega_c], \quad \zeta = [1, 3] \tag{17b}$$

The GO has then been compared with the extended Kalman filter (EKF), which is the most used estimator algorithm in the industry because of its straightforward construction [17]. Even if the EKF is usually applied to complete observability systems, in this investigation it has been used also when the choice of measurements leads to a rank deficient observability matrix. The EKF algorithm has been applied in the continuous form, reported in the following Equation (18).

$$\dot{\hat{x}}_i = \hat{f}_i(\hat{x}, u) + K_{EKF}(y - h(\hat{x})), x_{i0} = x_i(t_0) \tag{18a}$$

$$\dot{\hat{x}}_u = \hat{f}_u(\hat{x}, u), x_{u0} = x_u(t_0) \tag{18b}$$

$$K_{EKF} = P(t)H^T R^{-1} \tag{18c}$$

$$\dot{P}(t) = P(t)F(t) + F^T(t)P(t) + Q(t) - K_{EKF}HP, P(t_0) = P_0 \tag{18d}$$

$F(t)$ is the Jacobian of the vector field $\hat{f}_i$, calculated with respect to the innovated states, $P$ is the error covariance matrix of the innovated states, $H$ is the matrix of the derivative of the map $h$ with respect to the states, $Q$ and $R$ are, respectively, the covariance matrix of the model and measurements errors [9]. The constant matrix $Q$, $R$, and $P_0$ are tuning parameters of the estimation model and they have been calculated minimizing the error between the states calculated with the simulated plant and the estimator along a reference trajectory.

## 4. Results

### 4.1. Estimation Structure

The choice of the estimation structure has been carried out considering: (i) condition number and the minimum singular value of the Jacobian matrix $\Upsilon$ (or $\Upsilon_p$) for a different choice of measurements

and innovated states and (ii) evaluating the responses of the reconstructed states for a given trajectory. Temperature and dissolved oxygen measurements have always been considered available, according to the laboratory and industrial practice. On the other hand, sensors suited for ethanol measurements as well as substrate and biomass are not always available for large scale real-time applications [4]. According to the analysis reported in [3], two possible scenarios have been considered: (i) biomass concentration in the reactor is measured online or (ii) substrate concentration in the reactor is measured online. Using the representation in (9), the considered cases are reported in Equation (19):

$$y = \left( C_x, \ C_{O_2}, T_r, \ T_{ag} \right) \tag{19a}$$

$$y = \left( C_s, \ C_{O_2}, T_r, \ T_{ag} \right) \tag{19b}$$

where $y$ represents the measured output vector.

According to Equation (13), it is easy to demonstrate that no combination of indexes $\kappa_i$ satisfies the observability property for the output vector in Equation (19a). This implies that a full order observer is possible if the substrate concentration is measured online and therefore when using the output configuration reported in Equation (19b). In this case, the nonlinear estimation maps satisfying Equation (13a) are reported in Equation (20)

$$\Phi_1 = \left[ C_S, \ L_f C_S, \ C_{O_2}, \ L_f C_{O_2}, T_r, \ T_{ag} \right], \ \Phi_2 = \left[ C_S, \ L_f C_S, \ C_{O_2}, T_r, \ L_f C_{T_r}, T_{ag} \right] \tag{20}$$

A first comparison of the two structures can be carried out by considering the values of condition number and minimum singular value for the Jacobian of the maps (20), calculated by averaging along the trajectories obtained with input step changes T1 and T2 (Table 4) and reported in Table 5. The structure $\Phi_2$ seems to be more robust (lower condition number), but it shows a lower minimum singular value, indicating that changes in the states should affect the outputs to a lesser extent.

**Table 5.** Condition number and minimum singular value with four measurements.

|            | $\Phi_1$ | $\Phi_2$ |
| ---------- | -------- | -------- |
| $\Sigma$   | 1802.6   | 71.37    |
| $\underline{\sigma}$ | 0.086    | 0.03     |

The reconstruction capabilities of the two structures using the geometric observer are therefore calculated, using the input variations T1 and T2, reported in Table 4. Results are shown in Figures 1–4, only for the unmeasured variables, which are ethanol and biomass concentration. It is worth noticing that also the state values calculated only with the model used in the estimation algorithm (open-loop model), but without innovation are reported in order to better highlight the correction provided by the estimation algorithm.

It is possible to observe that using the map $\Phi_1$, allows a good reconstruction of the biomass behavior (Figure 1), while there is a large mismatch between the ethanol concentration obtained with the virtual plant and the reconstructed one (Figure 2).

When using the second configuration, results worsen, both for biomass (Figure 3) and for ethanol (Figure 4) concentration. It is worth noticing that the state's values estimated with map $\Phi_2$ are more corrupted by the measurement noise because in this case a greater observer gain has been used to decrease the offset.

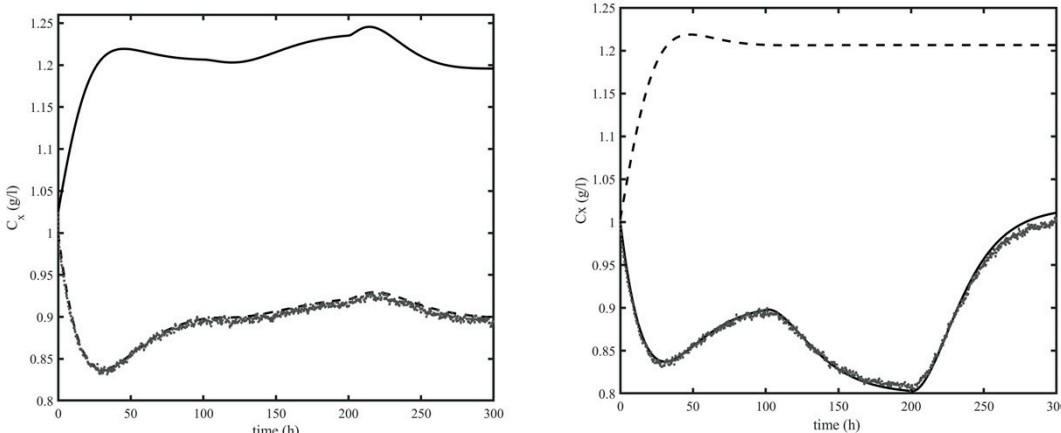

**Figure 1.** Dynamic response of biomass concentration calculated with the virtual plant (continuous line), open-loop model (dashed line) and geometric observer (GO) (dotted grey line) for structure $\phi_1$ along trajectory T1 (**left** panel) and T2 (**right** panel).

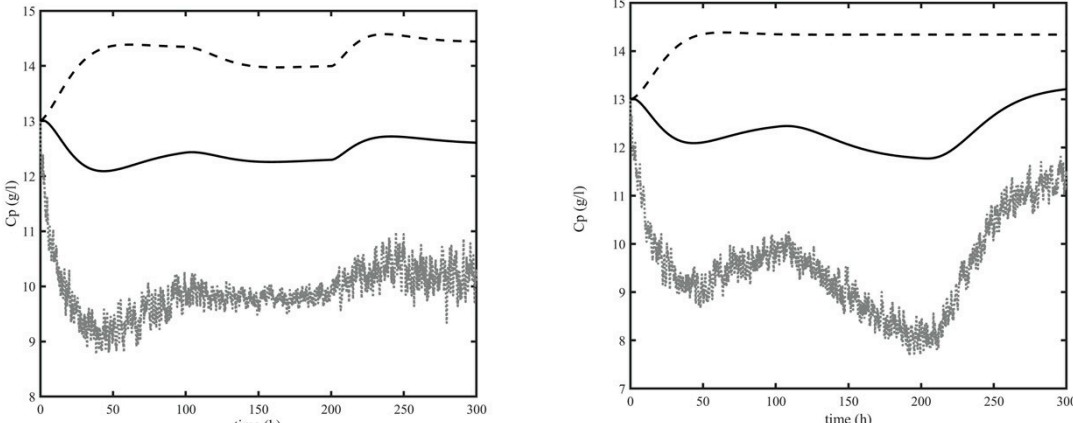

**Figure 2.** Dynamic response of ethanol concentration calculated with the virtual plant (continuous line), open-loop model (dashed line) and GO (dotted grey line) for structure $\phi_1$ along trajectory T1 (**left** panel) and T2 (**right** panel).

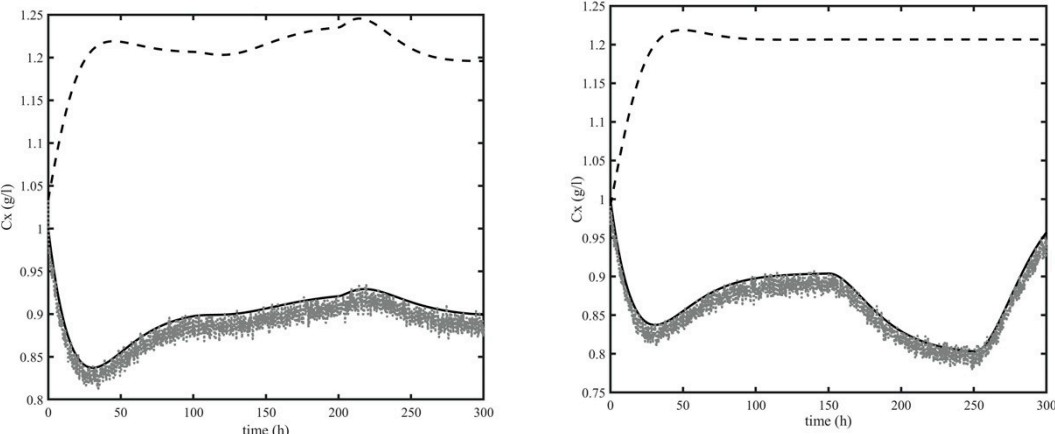

**Figure 3.** Dynamic response of biomass concentration calculated with the virtual plant (continuous line), open-loop model (dashed line) and GO (dotted grey line) for structure $\phi_2$ along trajectory T1 (**left** panel) and T2 (**right** panel).

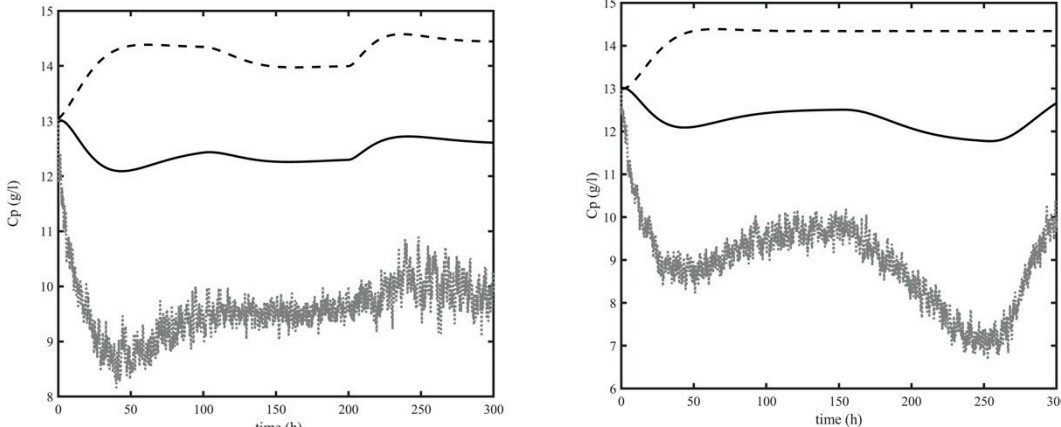

**Figure 4.** Dynamic response of ethanol concentration calculated with the virtual plant (continuous line), open-loop model (dashed line) and GO (dotted grey line) for structure $\phi_2$ along trajectory T1 (**left** panel) and T2 (**right** panel).

The two full order structures are not able to adequately estimate the product of the reactor, therefore a different solution is required to improve ethanol concentration. Using the same measured outputs, it is possible to improve estimation performance by reducing the order of the observer using only one Lie's derivative [22]. The maps reported in Equation (21) lead to five observable states and only one detectable.

$$\Phi_{p3} = \left[ C_s, \ C_{O_2}, T_r, \ L_f \ T_r, \ T_{ag} \right], \ \Phi_{p4} = \left[ C_s, \ L_f C_s, \ C_{O2}, \ T_r, \ T_{ag} \right] \qquad (21)$$

The rank of the Jacobian of the maps $\phi_{pi}$ ($i = 3, 4$) depends on the choice of the non innovated state ($\hat{x}_u$) between the two that are not measured, which are ethanol and biomass concentration. It can be verified that the map $\phi_{p3}$ can be inverted only if $C_x$ is innovated and $C_p$ is not. On the other hand, the Jacobian of map $\phi_{p4}$ always has a rank equal to five, regardless of the choice of the innovated states. Recalling Equation (15), the following partitions are considered:

$$x_i = \left[ C_x, C_s, \ C_{O_2}, T_r, \ T_{ag} \right], \ x_u = \left[ C_p \right] \qquad (22a)$$

$$x_i = \left[ C_p, C_s, \ C_{O_2}, T_r, \ T_{ag} \right] \qquad (22b)$$

The map $\phi_{p3}$ can be used with the partition in Equation (22a), while the map $\phi_{p4}$ can be used with both partitions in Equation (22a,b). Therefore, two different solutions are identified: $\phi_{p4,1}$ for partition (22b) and $\phi_{p4,2}$ for partition (22a). A first analysis of the possible configurations can be obtained by considering the minimum singular value and condition number reported in Table 6. The indexes' values are comparable; therefore, the evaluation of the best structure has been performed analyzing the reconstruction performance. Figures 5 and 6 represent the estimation of the unmeasured states (ethanol and biomass concentration) for the input step change T1 and T2 described in Table 4. The best reconstruction capabilities are shown by configuration $\phi_{p3}$ for both the states. This result may suggest that conditions calculated with Equation (14) are informative when the magnitude between the different configurations is significantly different, otherwise, it is necessary to evaluate the estimation capabilities by evaluating the estimator response for given input changes.

**Table 6.** Mean condition number and minimum singular value for low order structures.

|  | $\phi_{p3}$ ($C_P$ open loop) | $\phi_{p4,1}$ ($C_x$ open loop) | $\phi_{p4,2}$ ($C_P$ open loop) |
|---|---|---|---|
| $\Sigma$ | 2.15 | 8.75 | 1.53 |
| $\underline{\sigma}$ | 0.47 | 0.12 | 0.99 |

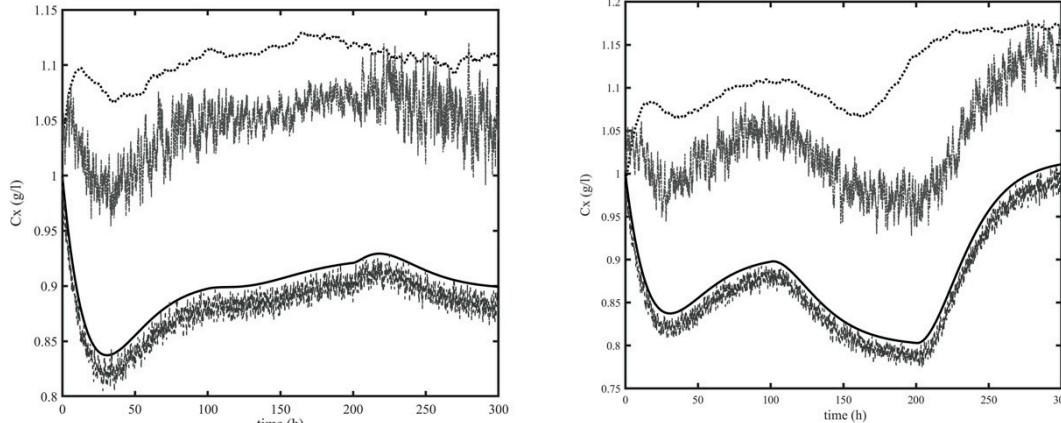

**Figure 5.** Dynamic response of the biomass concentration calculated with the virtual plant (continuous black line), GO with map $\phi_{p3}$ (dashed dark grey line), GO $\phi_{p4,1}$ (dotted black line), GO with map $\phi_{p4,2}$ (dashed-dotted grey line) along the trajectory T1 (**left** panel) and T2 (**right** panel).

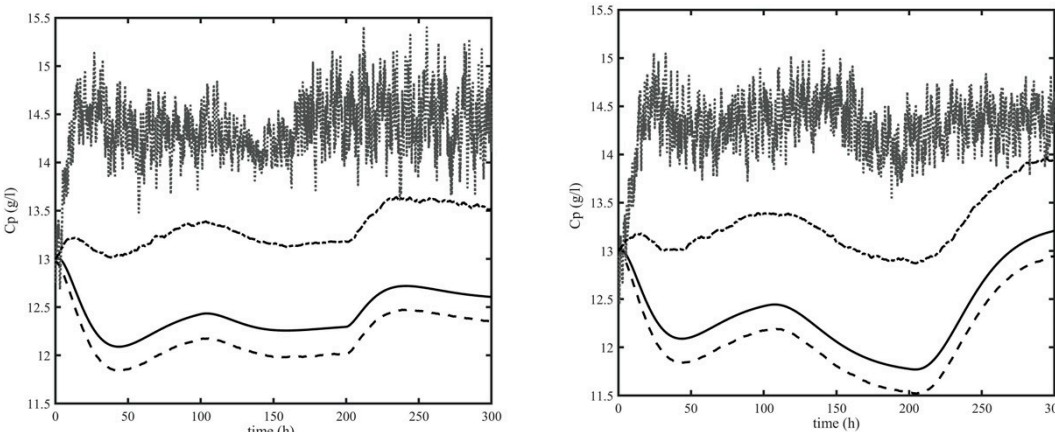

**Figure 6.** Dynamic response of the product concentration calculated with the virtual plant (continuous line), GO with map $\phi_{p3}$ (dashed black line), GO with map $\phi_{p4,1}$ (dotted black line), GO with map $\phi_{p4,2}$ (dashed-dotted grey line) along trajectory T1 (**left** panel) and T2 (**right** panel).

*4.2. Validation*

The analysis carried out in the previous section indicates the best estimation structure with four measured outputs. In order to validate the obtained results, a new test was carried out considering as reference trajectory the variation of the input temperature ($T_{in}$) as shown in Table 4 (Case T3). Figure 7 shows the dynamic behavior of biomass and product concentration and confirms that the proposed structure can effectively reconstruct the unmeasured states also with different process conditions. It is worth noticing that the ethanol concentration is not innovated, and the correction of the other states also has a positive impact on its estimation.

Using the same number and choice of measured outputs Equation (19b) and partition between innovated and not innovated states Equation (22a), the estimation task has been addressed using the extended Kalman filter (Figure 8). The main reason for using another algorithm as a measurement processor is to demonstrate that the estimator performance depends on the structure selection rather than estimation algorithm. EKF has been preferred for this validation because it is usually preferred in the industrial practice as it is easy to implement and robust if adequately calibrated [25,26].

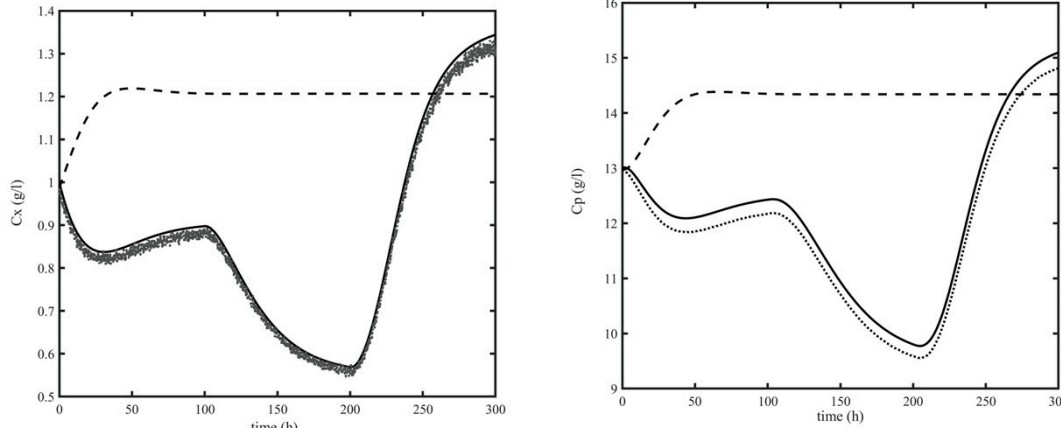

**Figure 7.** Dynamic response of biomass concentration (**left** panel) and ethanol concentration (**right** panel) calculated with the virtual plant (continuous line), open-loop model (dashed line) and GO (dotted grey line) for structure $\phi_{p3}$ along the trajectory T3.

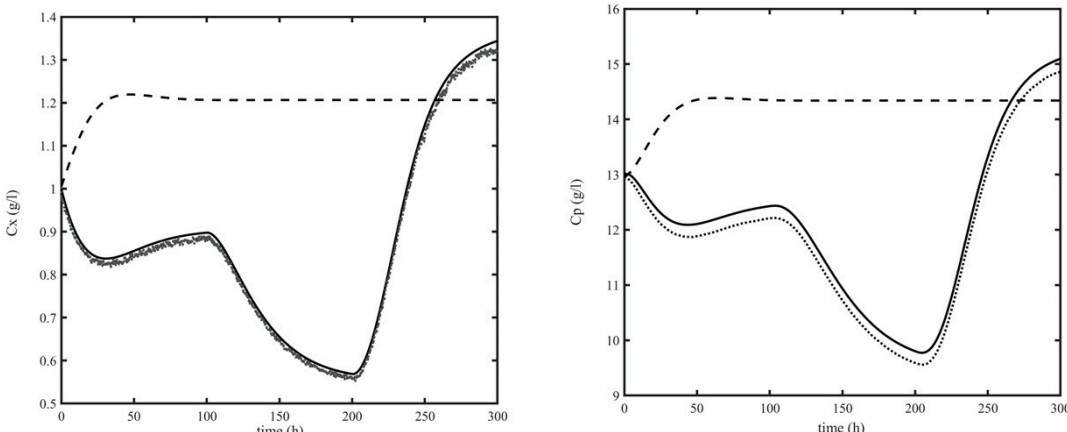

**Figure 8.** Dynamic response of biomass concentration and ethanol concentration calculated with the virtual plant (continuous line), open-loop model (dashed line) and extended Kalman filter (EKF) (dotted line) for structure $\phi_{p3}$ along the trajectory T3.

Results show that EKF can effectively reconstruct the unmeasured states, revealing that estimator structure design is the key step for a successful achievement of the estimation goals. The only difference between the two approaches is that the biomass calculated with the geometric observer is more affected by noise. This behavior can be explained by the presence of the Lie derivative in GO, which implies a higher sensitivity to measurement noise with respect to the EKF.

## 5. Conclusions

The problem of estimating unmeasured states in a bioreactor was addressed, and it was demonstrated that the estimation performance relies on an appropriate structure selection rather than the chosen measurement processor algorithm. An adjustable-structure geometric estimation approach was used, and the estimator structure constituted a design degree of freedom to improve its performance versus robustness behavior. The estimation structure design was based on estimability and detectability properties used together with a geometric approach. The analysis of the estimability measures showed the ill- and well-conditioned structures (condition number of the observability matrix), and the poorest estimation performance for the given structure (minimum singular value of the observability matrix). From the implementation stage with simulations, it was found that the results agreed with the ones obtained from the structural assessment when estimability measure values calculated for the different structures were significantly different. The used estimation algorithm

was the geometric observer with proportional innovation, which offers simplicity of tuning and implementation. With the aim of showing that the proposed procedure for choosing the estimation structure can be applied to other estimation techniques, the extended Kalman filter was also used as measurement processor algorithm. The obtained results showed that the two estimators lead to good estimation performance, with the only difference that the geometric observer estimation is more sensitive to measurement noise, probably because of the presence of the Lie derivative in the correction term. Summarizing, the systematic geometric approach led to the best solution for the estimation problem, giving a structure that did not depend on the correction algorithm. The latter can be chosen according to the wishes of the personnel of the plant or developer experience. It is worth noticing that the systematic tuning procedure of the geometric approach was very useful for comparing the reconstruction capabilities of the different structures. The results obtained in this paper in terms of methodology could be applied to more complex biotechnological processes, such as the obtainment of ethanol from cellulosic material, where the measurement devices for real-time application in the industry are still missing. In this case, the proposed approach can be used to detect the measurements that lead to the best reconstruction capabilities and invest in them.

**Author Contributions:** S.L. performed the analytic calculations and performed simulations; S.T. conceived the idea, proposed the computational model and wrote the paper; M.G. contributed to the analysis of the results and provided critical feedback; S.T. and M.G. reviewed the manuscript. All authors have read and agreed to the published version of the manuscript.

**Funding:** This research received no external funding.

**Conflicts of Interest:** The authors declare no conflict of interest.

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
