# Peer review of "A Geometric Observer-Assisted Approach to Tailor State Estimation in a Bioreactor for Ethanol Production"

_processes, doi:10.3390/pr8040480_

Round 1

Reviewer 1 Report

This work studied state estimation strategies in a bioreactor for ethanol production using the geometric observer (GO) and the extended Kalman filter (EKF). Different observer structures were studied under different operating conditions and the most promising GO structure was compared with the EKF.

The writing and presentation of the manuscript need to be improved. The reviewer found some undefined symbols and some equations which differ from the original model (Nagy, 2007) which should be corrected or explained its difference.

The plots presented are confusing. To use a grey scale might be helpful.

I do not recommend this paper in its current iteration.

My comments next:

Introduction

  1. I suggest to review the manuscript's title because it is too open and does not show the main contribution.
  2. I suggest the introduction to be written in three paragraphs. One introducing the topic (bioreactors), the second mentioning the need of state estimation strategies for process monitoring and the third mentioning clearly what is the contribution of this work.
  3. The introduction is written quite sloppy. Avoid using informal or out of context words/phrases, e.g., "reasonable prices", "Anyway", "the necessity to be more competitive", "lack of deep process understanding".
  4. Explain the further applications these soft-sensors could have for the process under study.
  5. Extend your literature review. Comment with more detail the benefits and limitations of using the EKF. In other words, why it is a good state observer but its construction does not tell which structure might be better when compared with the GO systematic approach. 

Process Model

  1. Erase the sentence in line 49: "It is indeed..."
  2. Check the model, eq. 1a (Ks), eq. 1e (+/- for last term).
  3. What criteria was used for varying the parameters among the virtual plant and estimator?
  4. Table 4. t=0h?

Estimation Problem

  1. Sentence starting in line 89: "State estimation technique..."
  2. individuation? isn't that word only for humans.
  3. Line 99. Erase "a more".
  4. What are the values or approximated values of the selected thresholds?
  5. Sentence in line 127: "The performance of an estimator is affected by..."
  6. Cite the source of sentence in line 159.
  7. Include additional references for the EKF.

Results

  1. Line 202 erase "very".
  2. Improve figures.
  3. Rewrite sentence in line 213-214 "The results get worst...".
  4. Erase in line 226 "which is, in general, the most important variable in this system".
  5. Rewrite sentence in line 240 "The obtained values...".
  6. Explain better the results in the validation section.

Conclusions

  1. I suggest to rewrite the conclusions stating the contribution of the study because people without the background would simply understand that the EKF is better.

Reviewer 2 Report

The authors describe a model based soft-sensor to estimate variables from other measured variables for potential use in continuous operation. The study is interesting but there are some minor issues that needs to be addressed.

  1. In the introduction the authors emphasize the advantages of continuous production and the inability of biotechnology processes to adopt the continuous process. But it is not clear how their methodology can address this issue. The authors can discuss how their method can be helpful for adopting continuous production for biotechnology processes.
  2. Figure 5 legend and presentation is not clear. Dashed-line and dashed-dotted lines are not visible. The authors should either change the appearance of the lines or mention which lines are overlapping and where they are.

Reviewer 3 Report

The paper under review considers the issue of an state estimation in a
bioreactor for ethanol production.

In the reviewer’s opinion, in general, the paper is quite interesting.
However, there are several important aspects that require authors comments
or possibly improvements:

1) All equations should be in the center of the page. The equation numbers should be on the right.
2) Abstract "In this work, a systematic approach based on the geometric observer is proposed to design " next in section 3.3
"two different algorithms have been selected and compared. The first one is the geometric observer [17]" then next, is section 5
"A procedure based on the geometric approach was used to address the estimation problem". The authors propose a new approach or
rather carry out a comparison of two methods???
3) The authors compare two extreme approaches. Geometric observer with Kalman observer. In geometric, the main problem is the
inverse of the observability matrix in each time step k, this is a problem from practical point of view. Please, explain deeply this issue.
4) Additional discussion regarding to implementation and possible numeric errors etc. is required.
5) Based on Comment 3 and all article the novelty presented in the article is not clear and not provided in clear form.

Round 2

Reviewer 1 Report

The authors have included the comments and corrections suggested by this reviewer. However, there are some minor corrections that I would like to suggest before publishing the manuscript.

Introduction:

Lines 30-31: Rewrite sentence “Model-driven soft sensors can be a possible approach to estimate variables from secondary measurements, to work as a backup when hardware sensors fail, and to perform fault detection.”

Line 36: Be consistent when using punctuation. I would suggest keep the format in line 102. Use first : then , only

Line 45: replace used with utilized, implemented, assessed, etc.

Line 47: Re write sentence “With the aim of showing that the proposed procedure for choosing the estimation structure can be applied to other estimation techniques, the extended Kalman filter (EKF) is also used as measurement processor algorithm.”

Process Model:

Line 52: Replace with: “…for the sake of clarity…”

Check equation 1a. All other equations appear to be bold except this one.

Line 69-71: Sentence “On the other hand,…” is too long. Rewrite it and separate it in 2 or 3 sentences.

Algorithms for the estimation problem:

Line 158: Rewrite “the coefficients of which are constant tuning parameters.”

Line 160: …dynamics predicted by the model are adjusted…

Line 167: Even if the EKF…

Results:

Line 186: erase “here”

Line 203: I would suggest replacing the word noting for noticing, observing, etc in the whole manuscript.

Figures 2 and 4, even if they appear to be in grey scale it is not as obvious as in Figure 3. Please check the intensity of the grey color.

Line 234: The maps reported in xx lead to… and only one detectable.

Line 269: noticing or observing.

Line 278: …rather than the estimation…

Line 280: for this idea, I would suggest to include a citation, e.g:

  • Leu, G., & Baratti, R. (2000). An extended Kalman filtering approach with a criterion to set its tuning parameters; application to a catalytic reactor. Computers & Chemical Engineering23(11-12), 1839-1849.
  • Salas, S. D., Ghadipasha, N., Zhu, W., Mcafee, T., Zekoski, T., Reed, W. F., & Romagnoli, J. A. (2018). Framework design for weight-average molecular weight control in semi-batch polymerization. Control Engineering Practice78, 12-23.

Author Response

The authors addressed all the reviewer’s comments. We acknowledge the reviewer for her/his fruitful considerations.   

Reviewer 3 Report

The paper under review considers the issue of a geometric observer assisted approach to tailor state estimation in a bioreactor for ethanol production

In the reviewer’s opinion, the answers to the question are satisfactory. What's more, the authors introduced a number of corrections to improve the article. In the reviewer's opinion, the article can be accepted for publication.

Author Response

We thank the reviewer for having appreciated our manuscript in the new form.